# Deep crypt secretory cells shape region-specific mucin glycosylation patterns in the mouse colon

**Daisuke Sugahara**[ORCID]*, **Hayato Kawakami, Yoshihiro Akimoto**

Laboratory of Microscopic Anatomy, Kyorin University School of Medicine, Mitaka, Tokyo, Japan

* d-sugahara@ks.kyorin-u.ac.jp

## Abstract

The colonic mucin layer, comprising highly glycosylated mucin proteins, is crucial for maintaining colonic health. Its region-specific glycosylation patterns are indispensable for adapting to distinct physiological and microbial environments along the colon, thus ensuring appropriate mucin layer function. However, the mechanisms underlying this region-specific glycosylation remain unknown. Here, using fluorescence-based immunohistological analyses of the colon from experimental mice, we demonstrated that along with contribution of goblet cells, as conventionally believed, mucin glycosylation involves deep crypt secretory (DCS) cells, a specialized mucin-producing cell population in the colon. Based on cKit/CD117 as a DCS cell marker, DCS and goblet cells are inversely distributed along the mouse colon: DCS cells predominate proximally, constituting nearly 70% of mucin-producing cells, whereas goblet cells are more abundant distally, indicating a dynamic shift in the predominant mucin-producing cell population along the colon. Immunofluorescence staining revealed that DCS cells produce distinctive mucin-glycans, including those with the Core3-glycan motif that exhibit region-specific distributions in the mucin layer. We found that the gradient distribution of DCS cells predominantly shapes their region-specific distribution, whereas the inverse distribution of goblet cells corresponds to the distal distribution of sulfated and sialylated glycans. Furthermore, the *in situ* Proximity Ligation Assay for specifically detecting Muc2 with distinct glycosylation, revealed that DCS and goblet cells produce different types of α1,2-fucosylated glycans on Muc2, indicating that the shift in the predominant mucin-producing cells drives region-specific α1,2-fucosylation on Muc2 across colonic regions. Although DCS cells are implicated in supporting the stem cell niche, their involvement in mucin production was unclear. We highlight the critical role of DCS cells in establishing regional glycosylation patterns. Our findings provide new insights into the cellular basis of mucin glycosylation, as well as their potential impact on colonic health and disease susceptibility in specific colonic regions.

**Data availability statement:** All relevant data are within the paper and its Supporting Information files.

**Funding:** D.S. was supported by Grants-in-Aid for Scientific Research (JSPS KAKENHI Grant No. 19K07275) from the Japan Society for the Promotion of Science (https://www.jsps.go.jp/english/e-grants/). The funder had no role in study design, data collection and analysis, decision to publish, or preparation of the manuscript.

**Competing interests:** The authors have declared that no competing interests exist.

## Introduction

The colonic epithelial surface is covered by a mucin layer serving as a physiological as well as immune barrier, protecting epithelial cells from harmful microbiota, their metabolites, and food digests [1–3]. This mucin layer predominantly comprises highly glycosylated mucin proteins, which constitute more than 80% of the mucin layer. A key feature of the mucin layer is its region-specific glycosylation patterns, which adapt to the distinct physiological and microbial environments across colonic regions, thereby ensuring appropriate barrier function [4–8]. These region-specific mucin glycosylation patterns play a key role in shaping a dynamic interface with the luminal environment, which varies along the colon length, in addition to ensuring barrier function. The distinct regional mucin-glycans contribute to establish and maintain symbiotic relationships with luminal microbes, leading to promotion of distinct microbial communities in different parts of the colon [9–11]. Accumulating evidence highlights the importance of region-dependent mucin glycosylation for colonic health [12–15]; however, the mechanisms underlying region-specific glycosylation remain poorly understood.

In the colon, goblet cells have been conventionally considered as solely responsible for producing mucin glycoproteins. Although goblet cells were once considered a homogeneous population, recent studies have demonstrated that they include multiple subpopulations with distinct characteristics, despite their similar appearance [16–18]. Deep crypt secretory (DCS) cells are a specialized secretory cell population in the colon [19]. Although DCS cells were initially considered a subpopulation of goblet cells owing to their expression of Muc2, a well-known goblet cell marker, recent single-cell RNA sequencing analyses suggest that DCS cells represent a distinct secretory cell population [20,21]. Despite ongoing debate over their classification, their unique gene expression profiles suggest specialized roles in colonic health maintenance. Although their functions remain unclear, DCS cells are implicated in maintaining the stem cell niche via the expression of epidermal growth factor and Notch ligands [22,23], and are suggested to regulate epithelial cell activity through the secretion of unique host defense peptides [24]. However, their specialized role in mucin production, an essential function of mucin-producing cells, remains poorly understood, highlighting the need for further investigation to better understand their unique contribution to colonic health. Notably, in the rat colon, DCS cells were distinguished from canonical goblet cells by their distinct mucin-glycan production [19]. Similarly, our previous study identified unique lectin reactivities in the secretory granules of mouse DCS cells, suggesting that distinct mucin-glycan production by DCS cells is common across species [25]. However, how these specific glycosylation features contribute to colonic health remains unexplored.

Therefore, this study aimed to elucidate the unique role of DCS cells in shaping the region-specific mucin glycosylation patterns along the colon. In this study, we investigated their distribution and characteristic glycan production across different regions of the mouse colon, based on the cell surface expression of the receptor tyrosine kinase cKit/CD117, as a marker for DCS cells [22]. Our results indicate that DCS cells contribute to shaping different mucin glycosylation patterns across colonic

regions, refining the conventional view that goblet cells are the sole contributors to mucin production. Overall, our findings provide novel insights into the cellular basis for region-specific mucin glycosylation.

## Materials and methods

### Mice and ethics statement

Male specific-pathogen-free C57BL/6JJcl mice were purchased from Japan Clea (Tokyo, Japan) at 8 weeks of age. Our animal research was conducted at the Institute of Laboratory Animals, Graduate School of Medicine, Kyorin University. All animal procedures were approved by the Committee on Animal Experimentation of Kyorin University (Approval Numbers 148 and 261) and were conducted in accordance with the institution's guidelines. All efforts were made to minimize animal suffering throughout the experimental process.

### Tissue preparation for immunostaining

Colon tissues were collected from the region immediately distal to the cecum to 1 cm proximal to the anus after sacrificing mice via $CO_2$ asphyxiation. The colon was divided into three equal-length segments representing the proximal, middle, and distal regions. The tissues were fixed in 4% paraformaldehyde at 4°C for 2 hours and cryoprotected in phosphate-buffered saline (PBS) containing 10% sucrose. Each tissue segment was opened longitudinally, feces were removed, then rolled and embedded in Optimal Cutting Temperature compound (Sakura Finetek, Tokyo, Japan). Cryostat sections measuring 8 µm in thickness were prepared and mounted on FRONTIER-coated glass slides (Matsunami Glass, Tokyo, Japan). Tissue sections were prepared from three individual mice.

### Immunofluorescence staining using antibodies and lectins

**Co-immunostaining for cKit and Muc2.** Air-dried frozen sections were washed with PBS and blocked with 4% (w/v) Block Ace (KAC Co., Kyoto, Japan) for 15 minutes at room temperature. The sections were incubated with rabbit anti-Muc2 polyclonal antibody (1:100; sc-15334, Santa Cruz Biotechnology, Santa Cruz, CA) for 2 hours at room temperature. After washing with PBS, the sections were incubated with goat anti-CD117/cKit polyclonal antibody (1:100; AF1356, R&D Systems, Minneapolis, MN) for 2 h at room temperature. Following another wash with PBS, the sections were incubated with a mixture of Alexa Fluor 488-conjugated donkey anti-goat IgG(H&L) polyclonal antibody (1:1200; ab150133, Abcam, Waltham, MA) and Alexa Fluor 555-conjugated donkey anti-rabbit IgG(H&L) polyclonal antibody (1:1200; ab150062, Abcam) for 1 hour at room temperature. Nuclear staining was performed using 4',6-diamidino-2-phenylindole (DAPI) (Dojindo Molecular Technologies, Kumamoto, Japan). The stained sections were mounted with Prolong Diamond Antifade Reagent (Life Technologies, Carlsbad, CA). Negative controls were prepared by omitting the primary antibodies to confirm the staining specificity.

**Fluorescent lectin staining.** Air-dried frozen sections were washed with PBS and blocked with 4% (w/v) Block Ace (KAC Co.) for 15 minutes at room temperature. The sections were incubated with biotinylated lectins (5 µg/ml) for 2 hours at room temperature. Recombinant N-terminal domain of the lectin BC2L-C derived from *Burkholderia cenocepacia* (BC2LCN) (FUJIFILM Wako Pure Chemical Corporation, Osaka, Japan) was biotinylated using a Biotin-labeling kit-NH$_2$ (Dojindo Molecular Technologies). The biotinylated lectins included *Griffonia simplicifolia* lectin II (GS-II), *Maackia amurensis* lectin I (MAL-I), *Maackia amurensis* lectin II (MAL-II), and *Ulex europaeus* agglutinin I (UEA-I), all obtained from Vector Laboratories (Burlingame, CA). After washing with PBS, the sections were incubated with goat anti-CD117/cKit polyclonal antibody (1:100; AF1356, R&D Systems) for 2 hours at room temperature. Following another wash with PBS, the sections were incubated with a mixture of Alexa Fluor 488-conjugated donkey anti-goat IgG(H&L) (1:1200; ab150133, Abcam) and Alexa Fluor 555-conjugated streptavidin (1:1200; S21381, Invitrogen, Carlsbad, CA) for 1 hour at room temperature. DAPI (Dojindo Molecular Technologies) was used for nuclear staining. The stained sections were mounted with Prolong Diamond Antifade Reagent (Life Technologies). Negative controls were prepared by omitting both the lectins and primary antibody to confirm staining specificity.

**Immunostaining for core3 β1,3-N-acetylglucosaminyltransferase 6 (C3GnT).** The Alexa Fluor 488 Tyramide SuperBoost Kit (B40943, Invitrogen) was used according to the manufacturer's instructions for detecting C3GnT. Air-dried frozen sections were washed with PBS and subjected to antigen retrieval by boiling in 10 mM Tris-HCl buffer (pH 9.0) containing 1 mM ethylenediaminetetraacetic acid disodium salt for 10 minutes. The sections were then washed with PBS and blocked with 4% (w/v) Block Ace (KAC Co.) for 15 minutes at room temperature. The sections were incubated with rabbit anti-B3GnT6 polyclonal antibody (1:200; HPA039805, Sigma-Aldrich Inc., St. Louis, MO) for 2 hours at room temperature. After washing with PBS, the sections were incubated with a pre-diluted poly-horseradish peroxidase-conjugated goat anti-rabbit IgG polyclonal antibody, provided by the manufacturer, for 1 hour at room temperature. Tyramide solution was applied for 5 minutes at room temperature, and the reaction was terminated by adding the provided reaction-stop reagent. The sections were then washed with PBS and incubated with biotinylated BC2LCN (5 µg/ml; FUJIFILM Wako Pure Chemical Corporation, prepared as described above) or biotinylated GS-II (5 µg/ml; Vector Laboratories) for 2 hours at room temperature. Finally, the sections were incubated with Alexa Fluor 555-conjugated streptavidin (1:1200; S21381, Invitrogen) for 1 hour at room temperature. After nuclear staining with DAPI (Dojindo Molecular Technologies), the sections were mounted with ProLong Diamond Antifade Reagent (Life Technologies). Negative controls were prepared by omitting the lectins and primary antibody to ensure staining specificity.

## Quantitative analysis of DCS cell distribution

The numbers of cKit-positive/Muc2-positive cells (cKit (+) cells) or Muc2-positive cells (Muc2 (+) cells) per crypt were quantified in sections of the proximal, middle, and distal colon. Well-oriented, full-length crypts were selected from sections co-immunostained for cKit and Muc2. In the middle colon, cell counts were performed separately for the anterior and posterior parts of each crypt. Tissue sections were prepared from three individual mice. The number of marker-positive cells was manually counted by visual inspection of the images, using consistent criteria throughout the study. More than 40 crypts were analyzed for each colon segment. Data are presented as the mean ± standard deviation. The number of cKit-negative/Muc2-positive cells (cKit (-) cells) was determined by subtracting the number of cKit (+) cells from the total number of Muc2 (+) cells. The percentage of cKit (+) or cKit (-) cells relative to the total mucin-producing cells (Muc2 (+) cells) was calculated.

## Statistical analysis

Statistical analyses were performed using Microsoft Excel (Microsoft, Redmond, WA). Statistical significance was evaluated using a two-tailed Welch's *t*-test, with $p < 0.05$ considered statistically significant.

### *In situ* Proximity Ligation Assay for Muc2 glycosylation

The *in situ* Proximity Ligation Assay (PLA) was performed using Duolink In Situ Detection Reagents Green (DUO92014, Sigma-Aldrich Inc.) according to the manufacturer's instructions. A combination of an anti-Muc2 antibody and a biotinylated lectin was used as the primary probe pair. The PLA oligonucleotide probes used were donkey anti-rabbit IgG polyclonal antibody conjugated with the PLA-Minus probe (DUO92005, Sigma-Aldrich Inc.) and Streptavidin conjugated with the PLA-Plus probe [26]. The PLA-Plus oligonucleotide probe was conjugated to Streptavidin (FUJIFILM Wako Pure Chemical Corporation) using Duolink *in situ* Probemaker (Sigma-Aldrich Inc.) following the manufacturer's protocol. Air-dried frozen sections were washed with PBS and blocked with 4% (w/v) Block Ace (KAC Co.) for 15 minutes at room temperature. The sections were incubated with biotinylated BC2LCN (10 µg/ml; FUJIFILM Wako Pure Chemical Corporation, prepared as described above) or UEA-I (10 µg/ml; Vector Laboratories) for 2 hours at room temperature. The sections were then washed with PBS and incubated with rabbit anti-MUC2 polyclonal antibody (1:40; sc-15334, Santa Cruz) for 16 hours at 4°C. After washing, the sections were incubated with a mixture of anti-rabbit IgG with the PLA-Minus oligonucleotide and Streptavidin conjugated with the PLA-Plus oligonucleotide in a humidified chamber for 60 minutes at 37°C.

The amplification reaction was performed by hybridizing and circularizing the oligonucleotides for 30 min at 37°C, followed by incubation with the manufacturer-supplied DNA polymerase at 37°C for 100 min in the dark to generate rolling circle amplification products. One of the PLA probe arms acted as a primer for the rolling circle amplification reaction, using the ligated circular DNA as a template. Fluorescence-labeled oligonucleotides were hybridized to the repeated sequences in the rolling circle amplification products. Sections were mounted with Prolong Diamond antifade reagent (Life Technologies) after nuclear staining with DAPI (Dojindo Molecular Technologies). The reaction specificity was examined by omitting the lectin and primary antibody. Results are representative of three independent experiments using tissue sections prepared from three individual mice.

### Fluorescent imaging using a confocal microscope

Fluorescence images were acquired using an LSM980 confocal microscope equipped with Zen 3.2 software (Carl Zeiss, Jena, Germany). Fluorescence images were processed using Adobe Photoshop (San Jose, CA) to adjust brightness and contrast. Identical Level Adjustment settings were uniformly applied to all images within each figure panel. These modifications were made exclusively for presentation purposes and did not affect any data analysis.

## Results

### The distribution of DCS cells along the colon length

**Histological analysis of DCS cell distribution.** We investigated the distribution of DCS cells along the length of the mouse colon by performing immunofluorescence co-staining on frozen tissue sections from the proximal, middle, and distal colon using anti-cKit and anti-Muc2 antibodies. In this study, DCS cells were classified as Muc2-positive cells expressing cKit, as described in previous studies [22], distinguishing them from canonical goblet cells, which also express Muc2 but not cKit. Previous studies on the distal colon have shown that cKit-labeled DCS cells are localized at the crypt base and are relatively sparse [22,25]. However, the existence of regional differences in their localization and abundance remained unclear.

The colonic tissues used exhibited typical histological features for each region, as described in the literature [27]. Specifically, the proximal colon displayed protruding mucosal folds with shallow crypts along their surface. The middle colon exhibited a non-folded epithelium with relatively deep crypts compared to those in other colonic regions, and the distal colon exhibited crypts with a noticeable tilt, likely influenced by the presence of longitudinal folds. In the proximal colon, cKit-positive/Muc2-positive cells (cKit (+) cells) were found in a significant portion of the crypts (Fig 1A, left panels), and were mainly distributed in the lower two-thirds of the crypt. In contrast, cKit-negative mucin-secreting cells (cKit-negative/ Muc2-positive cells; cKit (-) cells) were confined to the crypt entrance and luminal surface. In the middle colon, cKit (+) cells were still found in the lower half of the crypt (Fig 1A, middle panels), whereas the upper crypts contained cKit (-) cells. Toward the more distal parts of the colon, the frequency of cKit (-) cells increased, and they eventually became the predominant cell population in crypts (Fig 1A, right panels). Simultaneously, the frequency of cKit (+) cells decreased, with these cells remaining localized to the crypt base in the distal colon. Additionally, immunofluorescence analyses of the distal colon revealed that anti-Muc2 reactivity was less frequent in cKit (+) cells (Fig 1B, right panel). Secretory granules in these cells were also smaller compared to those in the cKit (-) cells. Notably, despite the proximal and distal colon having different embryonic origins [28], the distribution of DCS cells did not exhibit a clear boundary or dichotomous pattern reflecting these differing origins.

**Quantitative analysis of the DCS cell distribution.** Immunofluorescence analysis demonstrated that DCS cells are distributed throughout the colon, with their abundance varying along the colon length. To further characterize their distribution, we quantified the number of cKit (+) cells in well-oriented, full-length crypts from each colonic region. The middle colon was further subdivided into anterior and posterior parts to better capture the transition in cell distribution.

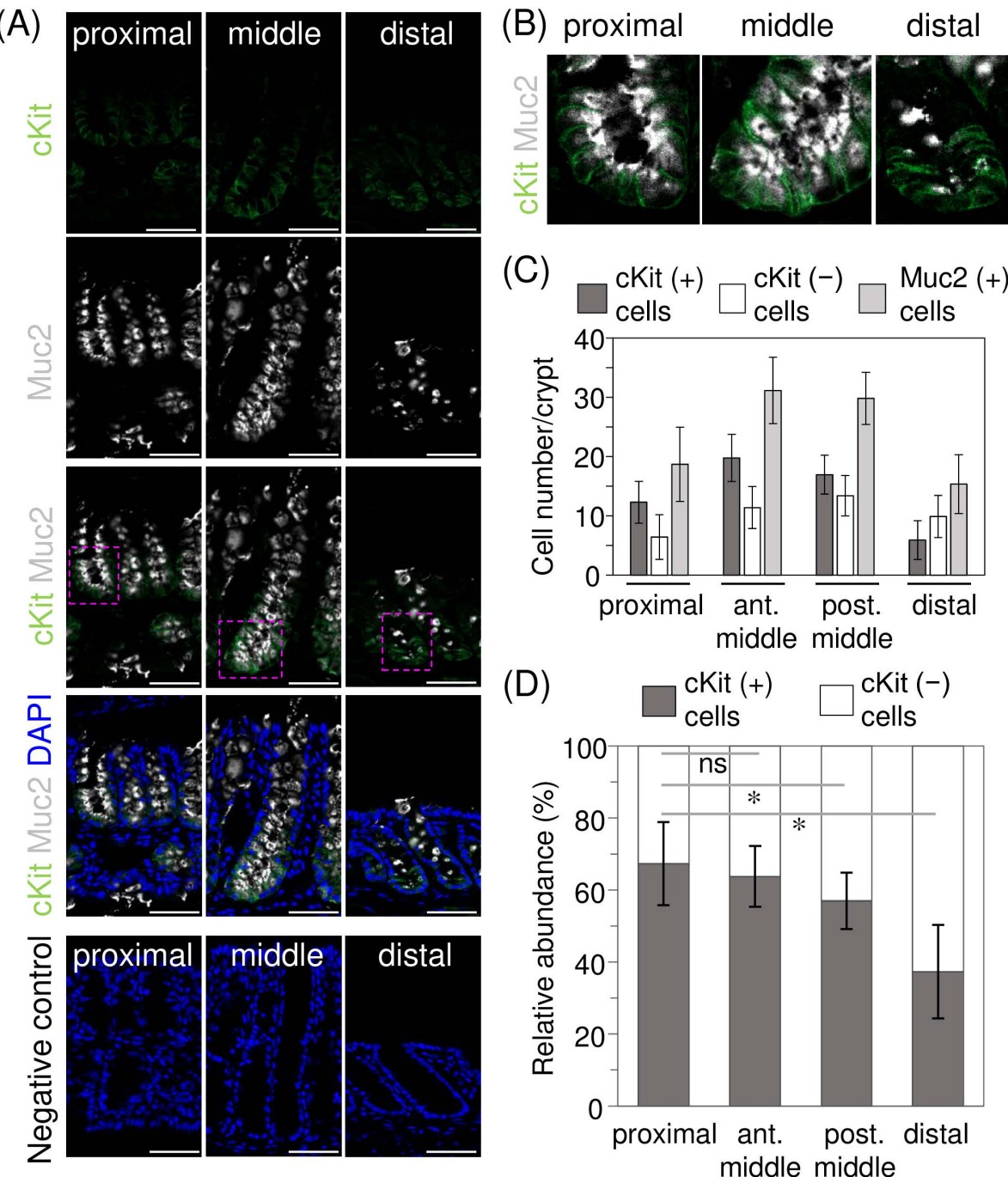

Fig 1. Deep crypt secretory (DCS) cells are predominantly distributed in the proximal colon, with a progressive decrease towards the distal colon. Co-immunostaining was performed using antibodies for cKit (DCS cells, green) and Muc2 (mucin-producing cells, white) to examine DCS cell distribution along the mouse colon. (A) Representative images of co-immunostaining for cKit (green) and Muc2 (white) in the proximal, middle, and distal colon. Enlarged views of the boxed regions are shown in (B). Nuclei were stained with DAPI (blue). Scale bar = 50 µm. No significant signal was detected in the negative control, where primary antibodies were omitted. Images are representative of three independent experiments for each colonic region. (B) Enlarged views of the crypt base from the boxed regions in (A), stained for cKit (green) and Muc2 (white). (C) Quantitative analysis of DCS cells in the co-immunostaining sections shown in Fig 1A. The number of cKit (+) cells or Muc2 (+) cells in the crypts was quantified in well-oriented, full-length crypts. The middle colon was further subdivided into anterior and posterior parts for more detailed analysis (labeled as "ant. middle" and "post. middle" in the figure). More than 40 crypts were analyzed for each segment. Data are presented as the mean ± standard deviation. (D) Relative abundance of DCS cells among Muc2-producing cells in different colonic regions. The percentage of cKit (+) or Kit (-) cells relative to the total Muc2 (+) mucin-producing

cells is shown. Data are presented as the mean ± standard deviation. A *p*-value was calculated by a two-tailed Welch's *t*-test. * indicates *p* < 0.001, and "ns" denotes not statistically significant (*p* = 0.071). The underlying numerical values used for the quantitative analyses in Figs 1C and 1D are provided in S1 Data. Abbreviations: DAPI, 4',6-diamidino-2-phenylindole; cKit (+) cells, cKit-positive/Muc2-positive cells; Kit (-) cells, cKit-negative/Muc2-positive cells; Muc2 (+) cells, Muc2-positive cells.

For comparison, the number of Muc2-positive cells (Muc2 (+) cells), representing all mucin-producing cells, was also quantified.

The quantitative analysis revealed that the anterior part of the middle colon contained the highest cell number of DCS cells, with 19.8 ± 4.0 cells/crypt (mean ± standard deviation), followed by the posterior part with 16.9 ± 3.3 cells/crypt (Fig 1C). The number of cKit (+) cells was lower in the proximal colon (12.3 ± 3.5 cells/crypt) and distal colon (5.9 ± 3.3 cells/crypt). The numerical values used for these analyses are available in S1 Data. In parallel, the numbers of Muc2 (+) cells also varied with the colonic region, with the highest abundance in the middle colon (31.1 ± 5.6 cells/crypt in the anterior part and 29.8 ± 4.4 cells/crypt in the posterior part), whereas lower numbers were observed in both the proximal (18.7 ± 6.3 cells/crypt) and distal colon (15.3 ± 5.0 cells/crypt). The higher cell numbers in the middle colon may correlate with increased crypt depth in this region (Fig 1A).

Subsequently, to assess the functional impact of DCS cells on mucin production, we examined the relative abundance of cKit (+) cells among mucin-secreting cells, independent of regional differences in crypt depth. The prevalence of cKit (+) and cKit (-) cells among Muc2 (+) cells were calculated based on the cell counts shown in Fig 1C. The highest prevalence of cKit (+) cells was observed in the proximal colon (67.3 ± 11.6%; mean ± standard deviation), despite their second-lowest cell number (Fig 1D). In the middle colon, the prevalence of cKit (+) cells decreased distally, with a significant reduction in DCS cell abundance in the posterior part (*p* < 0.001), but not in the anterior part (*p* = 0.071), compared to that in the proximal colon. Nevertheless, more than 50% of Muc2 (+) cells in the posterior part (57.0 ± 7.9%) remained cKit (+) cells. Eventually, the distal colon showed the lowest prevalence of cKit (+) cells (37.3 ± 13.0%).

Collectively, these results illustrate the changes in DCS cell prevalence along the colon, showing a progressive decrease from the proximal to the distal colon, particularly in the posterior middle and distal regions. Although DCS cells have been thought to localize at the crypt base, these results indicate that their positions within the crypt extend to the upper part depending on the colonic regions. Notably, DCS cells account for nearly 70% of mucin-secreting cells in the proximal regions of the colon, suggesting that DCS cells are the predominant mucin-producing cells in these regions.

### DCS cells produce distinctive mucin-glycans in the mouse colon

To investigate the implications of the proximal-to-distal gradient in DCS cell prevalence to mucin glycosylation, we examined the mucin-glycan production of DCS cells using a panel of lectins in fluorescence tissue staining. These lectins were selected to recognize representative glycan motifs relevant to mucin layer function.

**Expression of α1,2-fucosylated glycans recognized by BC2LCN in mucin-producing cells.** BC2LCN recognizes glycans with a terminal α1,2-fucosylated galactose (Fucα1,2Galβ1,3-) motif (Fuc, fucose; Gal, galactose) [29]. Our previous study identified the confined reactivity of BC2LCN in the secretory granules of cKit (+) cells in the distal colon [25]. However, its reactivity in other regions of the colon was unclear. In the proximal and middle colon, prominent BC2LCN reactivity was observed in the secretory granule of cKit (+) cells (Fig 2A, left and middle panels). In contrast, in the distal colon, BC2LCN reactivity was limited to a few cKit (+) cells (Fig 2A, right panel). Thus, we observed a progressive decline in the number of BC2LCN-positive cells from the proximal to the distal colon. Meanwhile, no reactivity was detected in cKit (-) cells throughout the colon, indicating that this glycan motif is specifically expressed by DCS cells.

**Expression of glycans recognized by GS-II in mucin-producing cells.** *Griffonia simplicifolia* lectin II (GS-II) recognizes a terminal β-linked *N*-acetylglucosamine (GlcNAc) residue [30]. Co-immunostaining with GS-II and

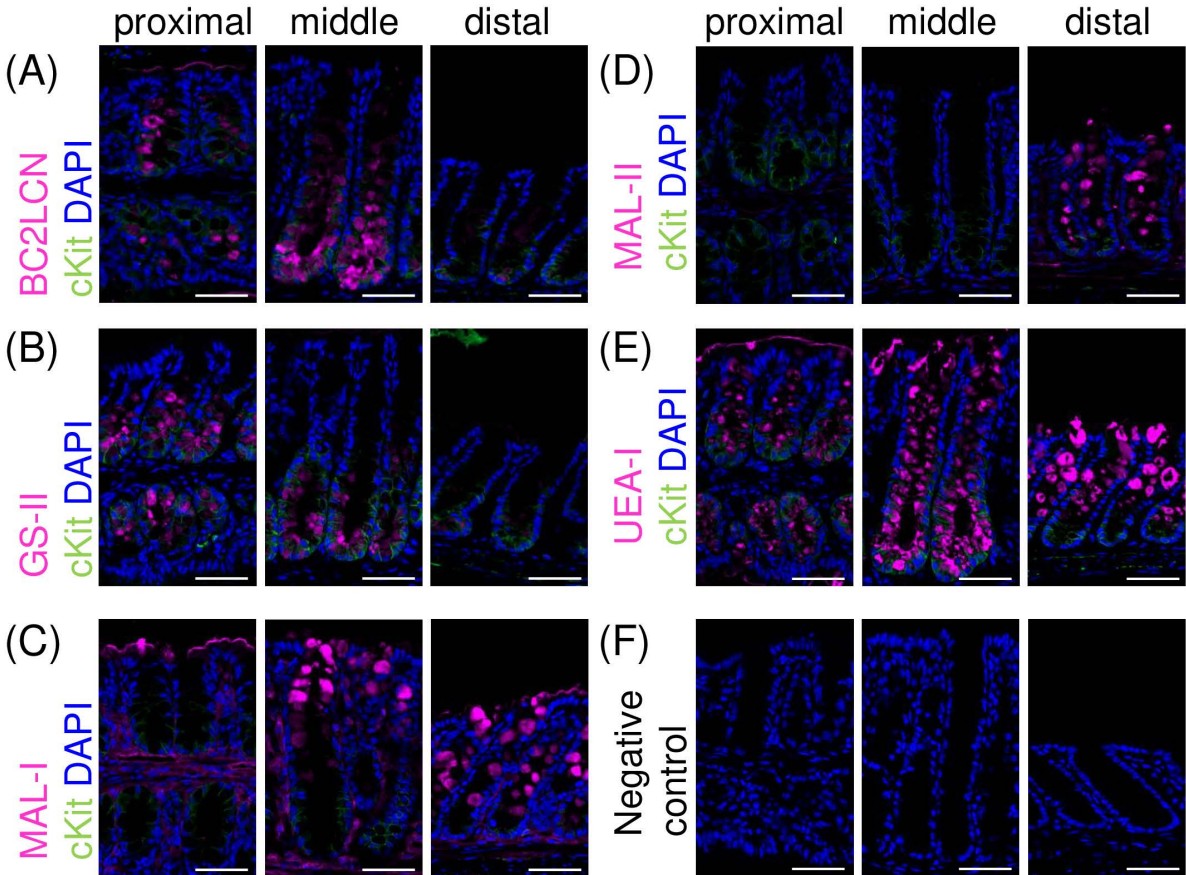

**Fig 2. DCS cells produce distinctive mucin-glycans in the mouse colon.** Co-immunostaining was performed using biotinylated lectins (magenta) and an antibody against cKit (green). The lectins used were BC2LCN, GS-II, MAL-I, MAL-II, and UEA-I. Nuclei were stained with DAPI (blue). Scale bar = 50 μm. Representative images from three independent experiments for each colonic region are shown. (A) BC2LCN: Recognizes glycans with a terminal α1,2-fucosylated galactose (Fucα1,2Galβ1,3-) motif. BC2LCN showed prominent reactivity in the secretory granules of cKit (+) cells in the proximal and middle colon. A few cKit (+) cells in the distal colon also exhibited BC2LCN reactivity. (B) GS-II: Recognizes glycans with a terminal β-linked GlcNAc residue. GS-II reactivity was prominent in cKit (+) cells in the proximal and middle colon, with a few cKit (+) cells in the distal colon also showing reactivity. (C) MAL-I: Recognizes glycans with a terminal Siaα2,3Galβ1,3GlcNAc- motif. MAL-I reactivity was prominent in the secretory granules of cKit (-) cells in the middle and distal colon. In the proximal colon, MAL-I reactivity was observed in cKit (-) cells at the luminal surface, but rarely in those within the crypts. MAL-I reactivity was hardly observed in cKit (+) cells throughout the colon. (D) MAL-II: Primarily recognizing sulfated glycans, but also Siaα2,3Galβ1,3GalNAc, known as the sialylated Core 1-glycan. MAL-II reactivity was more restricted compared to that of MAL-I, being confined to cKit (-) cells in the distal colon. (E) UEA-I: Recognizes glycans with a terminal Fucα1,2Galβ1,4- motif. In the proximal and middle colon, UEA-I reactivity was observed in both cKit (+) and cKit (-) cells. In the distal colon, UEA-I reactivity was confined to cKit (-) cells, but reactivity was hardly observed in cKit (+) cells. (F) Negative control: Staining performed without biotinylated lectins and anti-cKit antibody. No significant signal was detected. Abbreviations: BC2LCN, N-terminal domain of the lectin BC2L-C derived from *Burkholderia cenocepacia*; GS-II, *Griffonia simplicifolia* lectin II; MAL-I, *Maackia amurensis* lectin I; MAL-II, *Maackia amurensis* lectin II; UEA-I, *Ulex europaeus* agglutinin I; Fuc, fucose; Gal, galactose; GalNAc, *N*-acetylgalacosamine; GlcNAc, *N*-acetylglucosamine; Sia, sialic acid.

anti-cKit antibody revealed distinct lectin reactivity patterns between cKit (+) and cKit (-) cells, similar to those observed with BC2LCN staining. In the proximal and middle colon, GS-II reactivity was detected in the secretory granules of cKit (+) cells, but not in cKit (-) cells (Fig 2B, left and middle panels). In contrast, only a few cKit (+) cells in the distal colon exhibited GS-II reactivity (Fig 2B, right panel). Our findings further demonstrate that GS-II reactivity is specific to cKit (+) cells, and that its pattern, similar to that of BC2LCN (Fig 2A), decreases in frequency from the proximal to distal colon. These reactivities are consistent with a previous study showing GS-II reactivities in the lower crypts in the

proximal colon and the crypt bases of the distal colon [31]. However, these findings were not previously linked to DCS cells.

**Expression of sialylated glycans recognized by MAL-I in mucin-producing cells.** *Maackia amurensis* lectin I (MAL-I) recognizes glycans with a terminal α2,3-linked sialylated glycan (Siaα2,3Galβ1,4GlcNAc-) motif (Sia, sialic acid) [30]. In the proximal colon, MAL-I reactivity was observed in the secretory granules of cKit (-) cells at the luminal surface (Fig 2C, left panel). In the middle and distal colon, this reactivity was prominent in the secretory granules of cKit (-) cells in the upper crypt and at the luminal surface (Fig 2C, middle and right panels). In contrast, MAL-I reactivity was rarely observed in cKit (+) cells throughout the colon. These findings indicate that the glycans recognized by MAL-I are expressed in cKit (-) cells, but not in cKit (+) cells.

**Expression of acidic glycans recognized by MAL-II in mucin-producing cells.** *Maackia amurensis* lectin II (MAL-II) recognizes sialylated glycans, but its glycan-binding specificity differs from that of MAL-I. MAL-I binds to glycans with the Siaα2,3Galβ1,4GlcNAc- motif whereas MAL-II specifically recognizes Siaα2,3Galβ1,3GalNAc-, known as the sialylated Core1-glycan (GalNAc, *N*-acetylgalactosamine) [30]. However, in the mouse colon, MAL-II is primarily thought to bind sulfated glycans [15]. As expected, MAL-II exhibited a distinct binding pattern compared with that of MAL-I. In contrast to MAL-I, which reacted with cKit (-) cells throughout the colon, MAL-II reactivity was limited to cKit (-) cells in the distal colon (Fig 2D, right panel). In the middle and proximal colon, MAL-II reactivity was scarcely detected, even in cKit (-) cells (Fig 2D, left and middle panels). Similar to MAL-I, MAL-II reactivity was rarely observed in cKit (+) cells across all colonic regions. These results suggest that sulfated glycans, and potentially sialylated Core1-glycan, are specifically expressed in cKit (-) cells.

**Expression of α1,2-fucosylated glycans recognized by UEA-I in mucin-producing cells.** To further investigate α1,2-fucosylated glycans, we performed co-immunostaining using *Ulex europaeus* agglutinin I (UEA-I), which recognizes another α1,2-fucosylated glycan structure with a terminal Fucα1,2Galβ1,4- motif [30]. In contrast to BC2LCN reactivity, which was confined to cKit (+) cells, UEA-I reactivity was observed in both cKit (+) and cKit (-) cells in the proximal and middle colon (Fig 2E, left and middle panels). In contrast, in the distal colon, prominent UEA-I reactivity was confined to cKit (-) cells, with no reactivity observed in cKit (+) cells (Fig 2E, right panel). Overall, these results indicate that glycans with a terminal Fucα1,2Galβ1,4- motif are produced across all regions of the colon, except by distal cKit (+) cells.

Taken together, these findings underscore the distinct glycan production profiles of DCS cells along the colon. Consistent with our previous work showing a strong correlation between cKit expression and the distinctive mucin-glycan production by DCS cells [25], the present study further confirms that this characteristic is maintained in other regions of the colon. Specifically, BC2LCN and GS-II reactivities were observed exclusively in DCS cells, whereas MAL-I and MAL-II reactivities were confined to cKit (-) cells. Notably, the absence of MAL-I and MAL-II reactivity appears to be a distinguishing feature of DCS cells. Interestingly, a substantial subset of cKit (+) cells in the distal colon exhibited reduced or absent reactivity to all tested lectins, suggesting that distal DCS cells possess a distinct mucin glycosylation profile compared with those in more proximal regions. Although UEA-I reactivities were found in both DCS and cKit (-) cells, the protein backbones carrying these glycans differed between these cell populations, as elaborated in a later section.

## DCS cells specifically express C3GnT, a glycosyltransferase responsible for the biosynthesis of Core3-derived glycans

The Core3-glycan motif, GlcNAcβ1,3GalNAcα- directly linked to the serine/threonine residues of its protein backbone, represents a common glycan structure in the mouse colon. This motif serves as a core scaffold for extending various glycan structures at their terminal ends. Mucin-glycans elongated from this motif, referred to as Core3-derived glycans, show a well-defined, region-specific distribution within the mucin layer [12,14,15]. Consistently, core3 β1,3-*N*-acetylglucosaminyltransferase 6 (C3GnT), the glycosyltransferase solely responsible for Core3-derived glycan biosynthesis, shows high mRNA expression levels predominantly in the proximal and middle colon [14,15]. As Core3-derived glycans

often display distinctive terminal glycan motifs [32], the expression of C3GnT may contribute to the regional varieties in mucin glycosylation. However, the precise localization of C3GnT in specific cell populations remains unclear in mice, as well as in humans. Determining the localization of C3GnT will provide valuable insights into the mechanism underlying region-specific mucin glycosylation. In this context, we focused on the specific reactivity of GS-II in cKit (+) cells (Fig 2B), which recognizes the terminal β-linked GlcNAc residues, including those found on the Core3-glycan motif [30,33]. Based on this observation, we hypothesized that Core3-derived glycans are specifically produced by DCS cells. To verify this hypothesis, we examined the localization of C3GnT using immunofluorescence staining with an anti-C3GnT antibody, along with BC2LCN and GS-II to mark DCS cells with their specific glycan expression.

Co-immunostaining with BC2LCN and anti-C3GnT antibody showed that C3GnT was expressed in BC2LCN-positive mucin-producing cells in the proximal and middle colon (Fig 3A, left and middle panels). This co-expression indicates that DCS cells specifically express C3GnT. In these cells, the anti-C3GnT signal was localized in the peri-nuclear region, likely corresponding to the Golgi apparatus. This localization aligns with the expected location of C3GnT activity, supporting the validity of these findings. In contrast, in the distal colon, no C3GnT expression was detected in crypt epithelial cells, including those at the crypt base where cKit (+) cells are located (Fig 3A, right panel).

Similarly, co-immunostaining with GS-II and anti-C3GnT antibody revealed anti-C3GnT reactivity in GS-II-positive mucin-producing cells in the proximal and middle colon (Fig 3B, left and middle panels). Although GS-II can recognize a range of glycans with a terminal β-linked GlcNAc residue [30], co-expression of C3GnT supports its recognition of Core3-derived glycans in DCS cells. In contrast, although GS-II-positive cells were observed in the distal colon, these cells lacked anti-C3GnT reactivity, which suggests that GS-II binds to glycans with a terminal β-linked GlcNAc residue distinct from Core3-derived glycans.

Collectively, these findings demonstrate that C3GnT expression is specific to DCS cells, predominantly in the proximal and middle colon. This observation is consistent with previous quantitative PCR results demonstrating C3GnT mRNA expression in epithelial cells of these regions [14,15].

### *In situ* Proximity Ligation Assay reveals distinct Muc2 glycosylation patterns in DCS cells

The results shown in Figs 2 and 3 indicate that DCS cells produce distinctive glycans. To assess whether these glycans are attached to Muc2, which is the major component of the colonic mucus layer, we employed the *in situ* Proximity Ligation Assay (PLA) [34]. This immunohistochemical technique enables precise localization of specific glycans on target proteins [26,35–37]. Using this technique, we directly investigated whether DCS cells produce Muc2 with unique glycans. To aid in the interpretation of PLA results, we also conducted co-immunostaining using the same probe pairs (i.e., lectins and anti-Muc2 antibody) employed in the PLA experiments, as shown in S1 Fig.

***In situ* PLA of Muc2 glycosylation using BC2LCN.** To assess the presence of α1,2-fucosylated glycans recognized by BC2LCN (referred to as BC2LCN (+) glycans) on Muc2, we performed *in situ* PLA using an anti-Muc2 antibody and biotinylated BC2LCN. In the proximal colon, prominent PLA signals were found in a significant portion of the crypt, except at the crypt entrance and luminal surface (Fig 4A, left panel). In the middle colon, PLA signals were prominent from the crypt base to its middle (Fig 4A, middle panel). In contrast, in the distal colon, only sparse PLA signals were detected at the crypt base (Fig 4A, right panel). This observation is consistent with the limited presence of cKit (+) cells expressing BC2LCN (+) glycans in the distal colon (Fig 2A, right panel). No PLA signals were observed in regions dominated by cKit (-) cells, aligning with their lack of BC2LCN reactivity (Fig 2A). In co-immunostaining using BC2LCN and anti-Muc2 antibody, BC2LCN reactivity was distributed in the middle to lower regions of the crypts, where cKit (+) cells reside, and co-localized with Muc2 signals in secretory granules (S1A Fig). The spatial distribution of PLA signals obtained using the same probe pair closely matched these merged co-immunostaining signals. The spatial distribution of PLA signals obtained using the same probe pair closely matched these merged co-immunostaining signals. These results confirm that DCS cells exclusively produce mucins with BC2LCN (+) glycans, highlighting their unique mucin glycosylation profile.

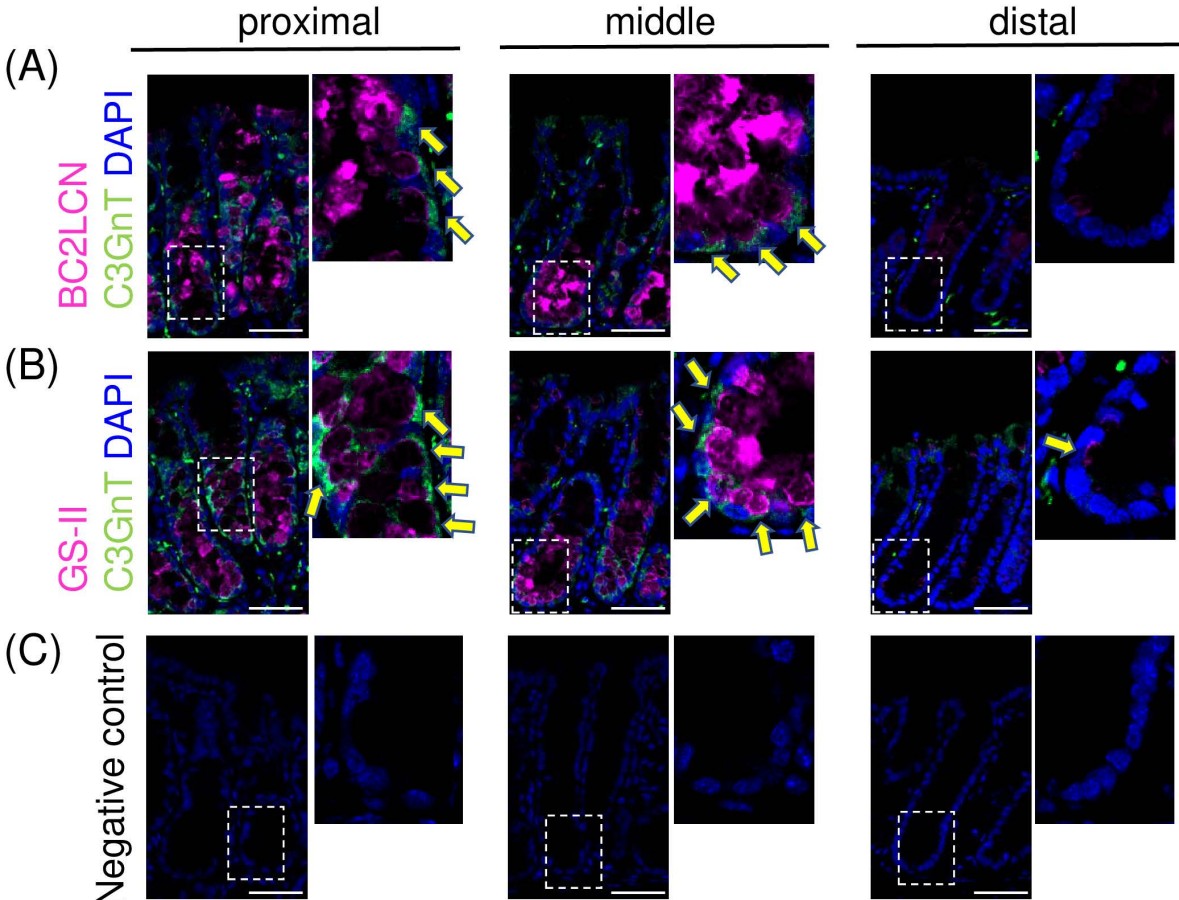

**Fig 3. The expression of C3GnT, a glycosyltransferase solely responsible for Core3-derived glycans, is confined to DCS cells.**
Co-immunostaining was performed using an antibody for core3 β1,3-N-acetylglucosaminyltransferase 6 (C3GnT) and biotinylated lectins (BC2LCN (A) or GS-II (B)). Enlarged views of the boxed regions are also shown. These lectins serve as markers for DCS cells, with GS-II additionally used to detect Core3-derived glycans. Nuclei are stained with DAPI (blue). Scale bar = 50 μm. Representative images from three independent experiments for each colonic region are shown. (A) Co-immunostaining for C3GnT (green) and BC2LCN (magenta). In the proximal and middle colon, C3GnT was localized to the peri-nuclear region of BC2LCN-positive cells (arrows), which were presumed to be DCS cells. No C3GnT reactivity was observed in the distal colon. (B) Co-immunostaining for C3GnT (green) and GS-II (magenta). C3GnT reactivity was observed in GS-II-positive cells in the proximal and middle colon (arrows). In the distal colon, GS-II reactivity was confined to a few epithelial cells at the crypt base (arrow), which did not show reactivity to the anti-C3GnT antibody. Antibody reactivity was also observed in stromal cells and luminal surface cells (Figs 3A and 3B). However, these reactivities were considered non-specific because they lacked supra-nuclear localization; this was further supported by the absence of GS-II reactivity in these cells. (C) Negative control staining with a biotinylated lectin and anti-C3GnT antibody omitted. No significant signal was detected.

Furthermore, the exclusive reactivity in cKit (+) cells and the spatially consistent signal distribution observed in the co-immunostaining experiment support the specificity and reliability of in situ PLA for detecting Muc2 glycosylation.

***In situ* PLA of Muc2 glycosylation using UEA-I.** Next, *in situ* PLA was performed using biotinylated UEA-I, which recognizes α1,2-fucosylated glycans distinct from BC2LCN (+) glycans [29,30]. Despite the distinctive glycan production of cKit (+) and cKit (-) cells (Figs 2A–2D), both populations were shown to produce α1,2-fucosylated glycans recognized by UEA-I (referred to as UEA-I (+) glycans) (Fig 2E). This prompted us to investigate whether any differences exist between these populations with respect to this glycan motif. Notably, although UEA-I reactivity was broadly observed in the mouse colon, PLA signals were more restricted in localization (Fig 4B). In the distal colon, prominent PLA signals were observed in the mid to upper crypts and luminal surface (Fig 4B, right panel), where cKit (-) cells predominantly reside (Fig 2E, right

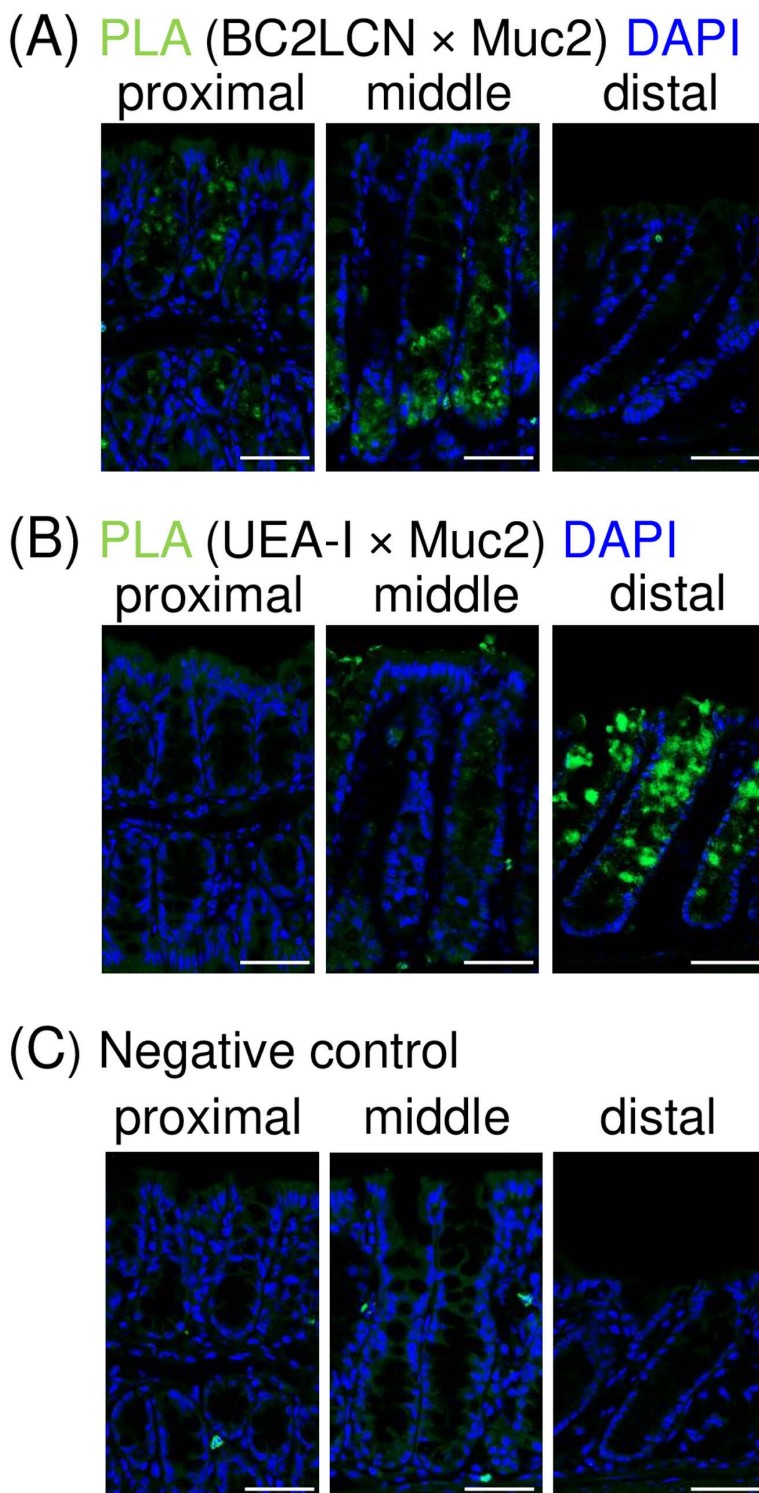

# (A) PLA (BC2LCN × Muc2) DAPI
### proximal  middle  distal

# (B) PLA (UEA-I × Muc2) DAPI
### proximal  middle  distal

# (C) Negative control
### proximal  middle  distal

**Fig 4.** ***In situ* Proximal Ligation Assay (PLA) reveals the unique Muc2 glycosylation of DCS cells.** *In situ* PLA was performed using a biotinylated lectin and an anti-Muc2 antibody. This technique visualizes the localization of Muc2 with specific glycan motifs on colonic tissue sections. The presence of PLA signals (green) indicates mucin-producing cell populations that produce Muc2 with the targeted glycan motif. Nuclei are stained with DAPI (blue). Scale bar = 50 μm. Data are representative of three independent experiments for each colonic region. (A) *In situ* PLA for detecting Muc2 with glycans terminated with the Fucα1,2Galβ1,3- motif. The assay used biotinylated BC2LCN and anti-Muc2 antibody. In the proximal and middle colon, prominent

PLA signals were detected at locations with cKit (+) cells. In the distal colon, only sparse PLA signals were detected at the crypt base. (B) *In situ* PLA for detecting Muc2 with glycans with a terminal Fucα1,2Galβ1,4- motif. The assay used biotinylated UEA-I and anti-Muc2 antibody. In the distal colon, prominent PLA signals were observed in the mid to upper crypts, where cKit (-) cells reside. In the middle colon, weak signals were observed in the upper crypts and luminal surface, gradually diminishing toward the proximal colon, where signals were scarcely detected. Notably, no signals were present at the crypt base, where cKit (+) cells reside, in the proximal and middle colon. (C) Negative control for *in situ* PLA with primary probes omitted. No significant PLA signal was detected in the secretory granules of epithelial cells. Some green signals on the epithelial cell surface and in the stromal tissue were considered non-specific, as neither anti-Muc2 nor lectin signals were observed in these regions in Figs 1A, 2A and 2E.

panel). Conversely, no PLA signals were observed at the crypt base, consistent with the lack of UEA-I reactivity in cKit (+) cells in the distal colon. Notably, no PLA signals were observed at the crypt base in the middle and proximal colon as well (Fig 4B, left and middle panels), despite the prominent UEA-I reactivity in the same regions (Fig 2E, left and middle panels). In contrast, as expected, weak PLA signals were detected in the upper crypts and luminal surface of the middle colon, becoming even fainter in the proximal colon. This weak reactivity aligns with the overall trend of UEA-I staining, which is less pronounced than in the distal colon.

Collectively, the *in situ* PLA showed that Muc2 with UEA-I (+) glycans is predominantly produced by cKit (-) cells. This result was unexpected, as UEA-I reactivity was also observed in cKit (+) cells in the proximal and middle colon (Fig 2E, left and middle panels). Supporting this observation, co-immunostaining using UEA-I and Muc2 showed that UEA-I signals were present in crypt regions where cKit (+) cells reside and co-localized with Muc2 signals (S1B Fig). This apparent discrepancy suggests that UEA-I (+) glycans in these regions are preferentially attached to mucin proteins other than Muc2. Therefore, our findings indicate that UEA-I (+) glycans are attached to distinct mucin proteins in DCS cells and cKit (-) cells.

## Discussion

In this study, we investigated the role of DCS cells in shaping region-specific mucin glycosylation patterns, which are essential for maintaining the colonic mucin layer. Although biochemical analyses have provided insights into the regional differences in mucin glycosylation [4–8], the mechanism shaping these variations remains poorly understood. Our findings suggest that DCS cells play a pivotal role in shaping these patterns through their unique glycan production and the regional differences in DCS cell prevalence.

### Unique roles of DCS cells in mucin glycosylation

A key finding of our study is the identification of a proximal-to-distal gradient in the distribution of DCS cells along the colon (Fig 1). Notably, our results indicate that DCS cells are the predominant source of proximal mucins. Previous studies have primarily focused on DCS cells in the distal colon, where their lower prevalence was believed to limit their contribution to mucin production [22,25]. Another important finding of this study is the distinctive glycan production by DCS cells (Fig 2). Importantly, we demonstrate that Core3-derived glycans are also exclusively produced by DCS cells, as evidenced by their specific expression of C3GnT (Fig 3). Our findings refine the current understanding of Core3-derived glycan expression, showing that it is limited to a specific mucin-producing cell population rather than being ubiquitous among all mucin-producing cell populations.

These findings advance our understanding of the unique role of DCS cells in mucin glycosylation. Biochemical studies have shown that Core3-derived glycans are abundant in the proximal mucin layer but barely detectable distally [12,14,15]. Our study demonstrates that this distribution pattern of Core3-derived glycans parallels the proximal-to-distal gradient of DCS cell distribution along the colon. Consequently, the distribution gradient of DCS cells is a key determinant of the regional distribution of Core3-derived glycans. Moreover, the expression of C3GnT in DCS cells may drive the production of glycans with unique structural features. Core3-derived glycans are thought to display terminal motifs influenced by the

action of glycosyltransferases that utilize the Core3-glycan motif as an efficient precursor [32]. Although our lectin staining did not allow complete characterization of these unique glycans, DCS cells may produce additional unique glycans beyond BC2LCN (+) glycans. Accordingly, the DCS cell distribution may determine the distribution of these DCS cell-specific glycans, similar to those of Core3-derived glycans [12,14,15].

## Cellular basis for region-specific mucin glycosylation

These findings provide novel insights into the cellular basis for region-specific mucin glycosylation. In the proximal regions of the colon, DCS cells are predominant contributors to mucin production, a conclusion further reinforced by the findings of Bergstrom et al. [15]. In their report, proximal mucins were characterized by the presence of Core3-derived glycans and the absence of glycans recognized by MAL-II, which are glycosylation patterns aligning with those produced by DCS cells (Figs 2 and 3). In contrast, mucin glycosylation in the distal colon is predominantly carried out by cKit (-) cells, representing canonical goblet cells. The quantitative analysis of cKit (-) cell distribution highlights their significant role in distal mucin glycosylation (Fig 1D). The lower mucin-producing activity of distal DCS cells (Fig 1B) further emphasizes the dominance of cKit (-) cells in mucin glycosylation in this region. Notably, cKit (-) cells specifically produce α2,3-sialylated and sulfated glycans recognized by MAL-I and MAL-II (Fig 2C and 2D). The distally predominant gradient of these cells parallels both the distribution of these acidic glycans in the mucin layer [15,38]. Consequently, the spatial distribution of cKit (-) cells along the colon also plays a pivotal role in shaping region-specific mucin glycosylation patterns, together with their unique glycan production.

By integrating these findings, this study refines the conventional view that goblet cells are the sole contributors to mucin glycosylation in the colon. A previously unappreciated shift in the predominant mucin-producing cell population along the colon likely underlies the distinct glycosylation patterns across colonic regions. The spatial arrangement of these cell populations, coupled with their unique glycan production profiles, likely plays a critical role in shaping region-specific mucin glycosylation. In this context, the regional glycan-profiles of the mucin layer are largely dictated by the relative abundance of these cell populations.

Notably, our results suggest that DCS cell prevalence alone does not completely explain the distribution of Core3-derived glycans. Although approximately 40% of mucin-producing cells in the distal colon are DCS cells (Fig 1D), Core3-derived glycans are rarely detected in the distal mucin layer [12,14,15]. This discrepancy may be attributed to the fact that distal DCS cells rarely express C3GnT (Fig 3). Thus, although the gradient distribution of DCS cells primarily shapes region-specific mucin glycosylation of Core3-derived glycans, regional differences in the glycan production profile of DCS cells may further enhance these patterns.

## Regional Muc2 glycosylation: Contributions of DCS and goblet cells

Building on these new findings, our results from *in situ* PLA further expand our understanding of the mechanisms underlying region-specific mucin glycosylation. This technique enabling the detection of Muc2 with specific glycans [26,35–37], highlights differences in the glycosylation profile of Muc2 between DCS and canonical goblet cells. Utilizing *in situ* PLA with BC2LCN and UEA-I, both of which recognize distinct types of α1,2-fucosylated glycans [29,30], we demonstrate that DCS and canonical goblet cells produce Muc2 with distinct α1,2-fucosylated glycans (Fig 4). Specifically, DCS cells exclusively produce Muc2 with BC2LCN (+) glycans, whereas cKit (-) cells exclusively produce Muc2 with UEA-I (+) glycans. The inverse distribution of DCS and cKit (-) cells likely drives region-specific α1,2-fucosylation on Muc2 across colonic regions.

Notably, DCS cells exhibited a unique ability to preferentially attach BC2LCN (+) glycans to Muc2, while UEA-I (+) glycans are rarely attached despite being produced (Figs 2E, 4B and S1B). This additional layer of complexity in the protein glycosylation process may contribute to reinforcing the regional specificity of α1,2-fucosylation on Muc2. The limited attachment of UEA-I (+) glycans by proximal and middle DCS cells results in preferential production of Muc2 with UEA-I

(+) glycans in more distal regions, primarily by canonical goblet cells. As Muc2 is the major component of the mucin layer, its glycosylation profiles likely shape the overall glycosylation patterns of the mucin layer. Consequently, these findings suggest that cell-type-specific Muc2 fucosylation is an additional factor contributing to regional Muc2 glycosylation, although its underlying mechanism remains unclear. As changes in fucosylation types on the mucin layer can affect microbial composition, region-specific α1,2-fucosylation patterns may significantly contribute to regional variations in intestinal microbial communities [39,40]. In conclusion, our findings indicate that mucin glycosylation have a more complex basis than previously thought, shaped by the coordinated contributions of both DCS cells and canonical goblet cells.

### Regulating the proximal-to-distal colon distribution of DCS cells

An important area for future research is to identify the mechanisms regulating DCS cell distribution along the proximal-to-distal colon. Our study suggests the importance of maintaining the cell abundance of mucin-producing cell populations for mucin layer function. The significance of maintaining these populations is also argued by a previous study showing that a reduction in a specific goblet cell subpopulation impairs the mucin barrier and increases susceptibility to colitis [41]. Based on these findings, our study raises the novel possibility that altered abundance of DCS cells could also disrupt mucin glycosylation patterns. As shown by Bergstrom et al., proximal mucin glycosylation is crucial for preventing colitis [15]. Thus, aberrant DCS cell abundance, particularly in the proximal colon, may compromise mucin layer integrity, potentially leading to colitis. This possibility underscores the need to explore the regulatory mechanisms governing DCS cell abundance.

Single-cell RNA sequencing analyses suggest that DCS cells differentiate from a common progenitor cell shared with canonical goblet cells [20,21]. In this context, the Wnt and Notch signaling pathways are particularly noteworthy as they have been implicated in DCS cell differentiation [22,23]. Regarding their fate determination, Sasaki et al. further suggest that simultaneous Notch inhibition and Wnt activation drive cells toward the DCS cell fate while suppressing goblet cell differentiation, as demonstrated using an *ex vivo* organoid model [23]. Regional variations in the balance of these pathways may determine DCS cell abundance in each colonic region, potentially affecting DCS cell distribution and, in turn, mucin glycosylation. Additionally, distal DCS cells exhibited lower mucin-producing activity (Fig 1B, right panel), suggesting that they are less mature as secretory cells than their counterparts in other regions. This immaturity may underlie their distinct mucin glycosylation profile, which is characterized by minimal reactivity to multiple lectins (Fig 2) and the absence of C3GnT expression (Fig 3). These findings may also support the link between the DCS cell development and regional mucin glycosylation patterns. Investigating whether these signaling pathways contribute to mucin glycosylation and their dysregulation leads to abnormalities in the mucin layer is important for further elucidating these mechanisms.

While intraepithelial signaling pathways likely play a crucial role, luminal environmental factors may also influence DCS cell abundance. Supporting this notion, a study by Mastrodonato et al. suggested that the proportion of mucin-producing cells with distinctive glycan production at the crypt base—likely corresponding to DCS cells—increased in response to a high-fat diet in mice [42]. This observation implies that changes in luminal cues, such as dietary components, may alter DCS cell abundance, either directly or indirectly. Such cues may also include shifts in the compositions of short-fatty acids and levels of lipopolysaccharides, which are associated with alterations in the intestinal microbiota and infections. Although our study did not investigate this aspect, investigating the environmental regulation of DCS cells will be crucial for understanding how regional mucin layer characteristics are maintained and how their disruption may contribute to disease.

### Conservation of DCS-like cells across species

A key limitation of this study is that our findings are based on mouse models, and it remains unclear whether a similar cellular mechanism underlies mucin glycosylation in humans and other vertebrates. Therefore, identifying potential counterparts of mouse DCS cells in other species is a crucial direction for future studies. In humans, given the remarkable

structural and functional similarities between the colon of human and rodent colons, it is plausible that mucin-producing cells with unique glycan profiles also exist. Indeed, multiple mucin-producing cell populations have already been identified in the human colon [16–18]. Furthermore, mucin-producing cells with distinctive glycan production have been reported in the lower crypt of the human colon [43,44]. Similarly, in the human colon, several glycosyltransferases exhibit region-dependent expression patterns [45–47], resembling the C3GnT distribution observed in the mouse colon. These regional variations may reflect differences in the abundance of a distinct mucin-producing cell population, likely corresponding to the human equivalent of mouse DCS cells. Supporting these observations, a recent study by Bustamante-Madrid et al. provided more direct evidence for the presence of DCS-like cells in the human colon [48]. Using transmission electron microscopy, they identified crypt-base epithelial cells with secretory granules—morphological features shared by DCS cells in mice and rats [19,22,23]. Notably, their study using human organoids also suggested that Notch signaling, previously implicated in the regulation of DCS cell abundance in mice [22,23], may similarly influence the development of these cells in humans.

Additionally, the recent study on the pig colon suggests the presence of DCS cells or their counterparts in other mammals [49]. Lectin staining in that study revealed Muc2-producing cells with distinct mucin glycosylation profiles located at the lower parts of the crypts, resembling the DCS cells described in our current study on mice. Together with recent findings in humans, these observations provide preliminary yet valuable evidence for the broader relevance of our findings to the human and other mammalian colons, and underscore the potential conservation of mechanisms regulating mucin glycosylation across species.

### Potential implications of human DCS cells in colonic diseases

From a pathophysiological perspective, advancing our understanding of human DCS cells in colonic diseases may offer new insights into the regionally distinct pathologies of colonic diseases. The altered prevalence of DCS cells maybe a contributing factor driving aberrant mucin glycosylation in patients with ulcerative colitis [50], alongside the previously assumed impact of their cellular dysfunction. Moreover, the distinctive characteristics of the distal DCS cells, such as their potential immaturity as secretory cells, could be associated with the distal-onset pattern commonly observed in this disease [51]. In addition to inflammatory diseases, these findings may shed light on the regional differences observed in colon cancer, where tumors in the proximal and distal colon differ in terms of epidemiology, histopathology, and patient outcomes [52]. A deeper understanding DCS cells in the human colon could thus serve as a foundation for developing novel preventive and therapeutic strategies for colonic diseases, such as ulcerative colitis and colon cancer.

## Supporting information

**S1 Data. Numerical values.** Data file including the numerical values used for the quantitative analysis of DCS cell distribution shown in Fig 1C and 1D.
(XLSX)

**S1 Fig. Co-immunostaining of Muc2 and lectins.** Murine colonic sections were stained with biotinylated BC2LCN or UEA-I (magenta) and anti-Muc2 antibody (green) following the procedures described in Materials and Methods, except that Alexa Fluor 488-conjugated donkey anti-rabbit IgG(H&L) polyclonal antibody (1:1200; ab150065, Abcam) was used as the secondary antibody for anti-Muc2 antibody. Nuclei were stained with DAPI (blue). Scale bar = 50 μm. Representative images from three independent experiments for each colonic region are shown. For clarity, representative crypts are marked with dotted lines. (A) Co-immunostaining of BC2LCN and Muc2. BC2LCN reactivity was confined to the middle to lower regions of the crypts, and co-localized with Muc2 signals. (B) Co-immunostaining of UEA-I and Muc2. UEA-I reactivity was localized to secretory granules and overlapped with Muc2 signals, including those in the middle to lower regions of

the crypts in the proximal and middle colon. (C) Negative control: Staining performed without biotinylated lectins and the anti-Muc2 antibody. No significant signal was detected.
(TIF)

## Acknowledgments

We would like to thank Dr. Akihiko Kudo and the members of the Laboratory of the Microscopic Anatomy, Kyorin University of School of the medicine, for their valuable input, encouragement, and for sharing the experimental resources. We also appreciate the support of the collaborative research facility at Kyorin University for animal care and access to experimental equipment.

## Author contributions

**Conceptualization:** Daisuke Sugahara.

**Data curation:** Daisuke Sugahara.

**Formal analysis:** Daisuke Sugahara.

**Funding acquisition:** Daisuke Sugahara.

**Investigation:** Daisuke Sugahara.

**Methodology:** Daisuke Sugahara, Hayato Kawakami, Yoshihiro Akimoto.

**Project administration:** Daisuke Sugahara.

**Resources:** Daisuke Sugahara, Hayato Kawakami, Yoshihiro Akimoto.

**Supervision:** Daisuke Sugahara.

**Validation:** Daisuke Sugahara, Hayato Kawakami, Yoshihiro Akimoto.

**Visualization:** Daisuke Sugahara.

**Writing – original draft:** Daisuke Sugahara.

**Writing – review & editing:** Daisuke Sugahara, Hayato Kawakami, Yoshihiro Akimoto.

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
