## [Decision Letter · Decision Letter 0]

PONE-D-25-10344Deep crypt secretory cells shape region-specific mucin glycosylation patterns in the mouse colon

PLOS ONE

Dear Dr. Sugahara,

Thank you for submitting your manuscript to PLOS ONE. The topic presented is both interesting and relevant, as the study addresses an important aspect of colon health: the glycosylation of mucins. The reviews are in general favourable and suggest that, subject to minor revisions, your paper could be suitable for publication.  Please consider these suggestions, and I look forward to receiving your revision.

We look forward to receiving your revised manuscript.

Kind regards,

Donatella Mentino

Academic Editor

PLOS ONE

Journal Requirements:

**Additional Editor Comments:**

The topic presented is both interesting and relevant, as the study addresses an important aspect of colon health: the glycosylation of mucins. This research identifies a new key player, DCS cells, which could have significant implications for our understanding of mucin biology and their role in colon health. By exploring the functions and mechanisms of these cells, the study opens new avenues for research that could lead to innovative therapeutic strategies and a deeper understanding of the complexities of mucin interactions within the colonic environment. This contribution is fundamental for advancing our knowledge and addressing related health issues.

Areas for improvement include: Clarity of Terminology: It is essential to clearly define technical terms and abbreviations used in the manuscript to enhance understanding for a broader audience. References: Ensure that all citations are accurate and complete and that they are correctly referenced in the text. Linguistic Review: Conduct a thorough linguistic review of the manuscript to correct any grammatical or spelling errors.

Several points need clarification and integration into the text: Has the expression of DCS been observed in other vertebrates (e.g., Balestra)? A comparison could be made. This could serve as a valuable starting point for further research to investigate the mechanisms regulating glycosylation among different vertebrate species. It would be possible to build on the data already published and integrate them into the text. Does DCS expression change under a variety of experimental conditions? Recent literature regarding alterations in the binding of glycosidic residues in the colons of mice fed a high-fat diet could be included, along with conducting an analysis of DCS expression.

Reviewers' comments:

Reviewer's Responses to Questions

**Comments to the Author**

1. Is the manuscript technically sound, and do the data support the conclusions?

Reviewer #1: Yes

Reviewer #2: Partly

2. Has the statistical analysis been performed appropriately and rigorously? 

Reviewer #1: I Don't Know

Reviewer #2: Yes

3. Have the authors made all data underlying the findings in their manuscript fully available?

Reviewer #1: No

Reviewer #2: Yes

4. Is the manuscript presented in an intelligible fashion and written in standard English?

Reviewer #1: Yes

Reviewer #2: Yes

5. Review Comments to the Author

Reviewer #1: My opinion is that the findings in this original research article fit the scope of PLOS One and are scientifically sound and novel.

However, to my knowledge Welch’s t-test assumes normal distribution of data but can be valid with a symmetric, non-skewed, sample size of more than 50 samples (https://doi.org/10.1177/0004563221992088). How was the data tested for normal distribution, or did it meet other criteria for Welch’s t-test to be used that I’m not aware of?

Can you supply the data underlying the quantitative analysis of the DCS cell distribution as supplementary information or deposited to a clearly stated public repository to meet the PLOS One data availability requirements?

Reviewer #2: This is a nicely written manuscript reporting the gradient distribution of DCS cells along the colon and the underlying role of region specific glycosylation in colon health. The authors elaborate on previously reported research to determine correlation of Core3 glycosylation and proximal mucins, and provide scientific evidence linking it to DCS cell distribution. I have a few minor comments and questions as follows-

1. Since this work is heavily relying on immunostaining, could the authors discuss if an isotope control was included in addition to the negative control, to confirm specificity of primary antibody? This is specifically, in reference to figure 4 (showing the UEA-1 reactivity in the in situ PLA).

2. I am also curious why Adobe photoshop was used for image analysis, instead of other open source software available for scientific image analysis.

3. Other than mucin production, are there any more functional differences expected in the DCS cells distributed along the proximal, middle and distal colon?

4. Line 620-624 discuss the variations in the Notch and Wnt signaling pathways (presumable originating from Lgr+ stem cells) that could drive the differences in DCS distribution and glycosylation. Since Notch and Wnt are highly conserved across species, can the authors comment if this has been studied in human organoid/stem cell models or if they plan to? This could help hypothesize similar regional glycosylation differences and help address the limitations (in line 640-641) of extending this work to human DCSs.

6. PLOS authors have the option to publish the peer review history of their article (what does this mean? ). If published, this will include your full peer review and any attached files.

**Do you want your identity to be public for this peer review?** For information about this choice, including consent withdrawal, please see our Privacy Policy .

Reviewer #1: **Yes: ** Joonas Terävä

Reviewer #2: No

---

## [Author Response · Author response to Decision Letter 1]

13 May 2025

Dear Academic Editor and Reviewers,

We would like to sincerely thank the Academic Editor and Reviewers for their careful evaluation of our manuscript and for providing thoughtful and constructive comments to improve its clarity and quality.

In response, we have revised the manuscript accordingly. Below, we provide point-by-point responses to all comments from the Academic Editor and Reviewers. A summary of the changes made is as follows:

- All revisions in the manuscript are highlighted in blue.

- Newly generated Supporting Information files have been added:

S1 Data (XLSX)

S1 Fig (TIF)

- Reference numbering has been corrected:

(i) Three new references (#42, #48, and #49) have been added.

(ii) The original references #42–49 have been renumbered accordingly.

(iii) A citation error in the original manuscript at line 611 has been corrected: Nyström et al. is now properly cited as reference #41 (previously #40).

- These changes do not affect the interpretation of the results or the overall conclusions of the study.

We believe that the revised manuscript fully addresses the concerns raised, and we respectfully submit it for your further consideration.

Sincerely,

Daisuke Sugahara

(on behalf of all authors)

Response to Academic Editor

General Comment

The topic presented is both interesting and relevant, as the study addresses an important aspect of colon health: the glycosylation of mucins. This research identifies a new key player, DCS cells, which could have significant implications for our understanding of mucin biology and their role in colon health. By exploring the functions and mechanisms of these cells, the study opens new avenues for research that could lead to innovative therapeutic strategies and a deeper understanding of the complexities of mucin interactions within the colonic environment. This contribution is fundamental for advancing our knowledge and addressing related health issues.

Response

We appreciate the Editor’s thoughtful and encouraging comments on the significance of our study. We are pleased that the potential relevance of DCS cells to mucin biology and colon health was recognized, and we value your perspective on how this work may contribute to a deeper understanding of region-specific glycosylation and mucosal function. The editor’s feedback was very helpful in clarifying the broader implications of our findings in the revised manuscript.

Comment (1)

Has the expression of DCS been observed in other vertebrates (e.g., Balestra)? A comparison could be made. This could serve as a valuable starting point for further research to investigate the mechanisms regulating glycosylation among different vertebrate species. It would be possible to build on the data already published and integrate them into the text. Does DCS expression change under a variety of experimental conditions? Recent literature regarding alterations in the binding of glycosidic residues in the colons of mice fed a high-fat diet could be included, along with conducting an analysis of DCS expression.

Response

We sincerely appreciate the valuable comments. We understand that the comments mainly concern the following two points:

(i) Whether the presence of DCS cells has been observed in other vertebrate species?

(ii) Whether DCS cell distribution or expression changes under different experimental conditions?

Below, we provide our responses to each point.

(i) We appreciate the insightful suggestion on the presence of DCS cells in other vertebrates, which is relevant to potential for the presence of a similar mechanism regulating mucin glycosylation across species. To date, there have been no reports directly indicating the presence of DCS cells or their equivalents in vertebrates other than mice and rats. DCS cells are considered counterparts analogous to the Paneth cells in the small intestine. While Paneth cells exhibit differences in morphology and distribution among animal species, they are known to exist across a broad range of mammals, including pigs, horses, rabbits, and cattle, as well as in chickens (reviewed in Cui C et al. J Anim Sci Biotechnol. 2023;14(1):118.). Although their apparent characteristics may differ from those of mice and rats, it is plausible that DCS cells, or analogous cell types, are also present in a wide variety of mammals. Among these, the spiral colon of pigs may be the most promising candidate. In a study by Lin SJ et al. (Front Cell Infect Microbiol. 2023;12:1042815.), lectin staining of pig colon tissue suggested the presence of Muc2-producing cells with distinct mucin glycosylation profiles at the lower parts of the crypts, resembling the DCS cells described in our current study.

Even more important from the perspective of maintaining human health and preventing disease, a recent study by Bustamante-Madrid P et al. (Cell Death Dis. 2024;15(4):301.) suggests the existence of DCS-like cells in health human colon. As we also mention in our reply to Reviewer#2’s comment (4), their study further indicated that Notch singling may be involved in the differentiation of these cells, consistent with previous findings in mice. However, further studies are needed to determine whether these DCS-like cells in pigs and humans are functionally equivalent to mouse DCS cells.

Considering the potential significance of DCS-like cells across species, we have added a new subsection to the Discussion section entitled “Conservation of DCS-like cells across species” (Line 660-685). In this new section, we cited their work and incorporate relevant descriptions (Line 673 and 679). Accordingly, the subsection “Potential implications of human DCS cells in colonic diseases” has been shortened, as some of its content was moved to the new subsection, and minor revisions have also been made to improve clarity and flow.

(ii) We also appreciate the Editor’s suggestion regarding the inclusion of a recent study that investigates changes in glycan expression patterns under different experimental conditions such as high-fat diet. As the Editor points out, we also consider that the study by Mastridonato M et al. (Histochem Cell Biol. 2014;142(4):449-59.) indicates changes in DCS cell distribution under a particular experimental condition in the mouse colon. Their histochemical analysis revealed the presence of mucin-producing cells with distinct mucin glycosylation profiles located at the lower parts of the crypts. Although Mastridonato et al. did not specifically identify these cell types, we believe that these cells correspond to DCS cells. Notably, their results showed that the crypt regions occupied by these distinctive cells expanded in mice fed a high-fat diet, suggesting that the distribution (abundance) of DCS cells changes in response to dietary components.

Their study also indicated that mucin-producing cells in the proximal colon, likely including DCS cells, produce acidic glycans. This observation differs from our findings and those of Bergstrom K et al. (Science. 2020;370(6515):467-72.). Although the reason for this discrepancy remain unclear, their results may suggest that this cell population is responsive to dietary or environmental cues. Accordingly, we have therefore cited this study and briefly discussed it in the Discussion section of the revised manuscript (Line 649-659).

Additionally, the experimental results reported by Lin et al., cited above, may also suggest that the area occupied by DCS-like cells expands in response to microbial infection in the pig colon. These findings, together with those of Mastridonato et al., support our notion that DCS cells play crucial and unique roles in mucin production and the maintenance of colonic health. However, as we are not sufficiently familiar with the histological characteristics of the porcine large intestine, and since alternative interpretations were presented in their article, we have opted not to include this point in the revised manuscript. We nevertheless appreciate the opportunity to consider this perspective.

Response to Reviewer#1

General Comment

My opinion is that the findings in this original research article fit the scope of PLOS One and are scientifically sound and novel.

Response

We sincerely appreciate the reviewer’s positive evaluation, especially regarding the scientific soundness and novelty of our findings.

Comment (1)

However, to my knowledge Welch’s t-test assumes normal distribution of data but can be valid with a symmetric, non-skewed, sample size of more than 50 samples (https://doi.org/10.1177/0004563221992088). How was the data tested for normal distribution, or did it meet other criteria for Welch’s t-test to be used that I’m not aware of?

Can you supply the data underlying the quantitative analysis of the DCS cell distribution as supplementary information or deposited to a clearly stated public repository to meet the PLOS One data availability requirements?

Response

We appreciate the reviewer's comment on this point. In response to the reviewer’s suggestion, we have added the numerical data underlying the quantitative analysis of DCS cell distribution (Figs 1C and 1D) as Supporting Information (S1 Data). We have also referred to this addition in the revised manuscript (Line 254-255 and 270).

We believe that our data meet the assumptions for Welch’s t-test, as outlined in the literature cited in the reviewer’s comment (West RM. Ann Clin Biochem. 2021;58(4):267-69.). It has been suggested that if the mean of a dataset is more than twice its standard deviation, the distribution can be considered approximately normal or not strongly skewed. As shown in the table below (also included in S1 Data for transparency), the mean exceeded twice the standard deviation in all colonic regions, suggesting relatively low skewness in our data.

Colonic region Sample size (n) Mean of DCS cell abundance (%) Standard deviation (SD) 2 xSD Mean > 2 x SD? Normal distribution?

Proximal 68 67.3 11.6 23.1 Yes Yes/approximately

Anterior Middle 42 63.8 8.4 16.9 Yes Yes/approximately

Posterior Middle 55 57.0 7.9 15.7 Yes Yes/approximately

Distal 109 37.3 13.0 25.9 Yes Yes/approximately

Another important assumption for Welch’s t-test is that the sample size is sufficiently large; in practice, a sample size of n > 50 is often considered adequate. In our dataset, the smallest group size was n = 42 (from the anterior region of the middle colon), which is not substantially smaller than this threshold. Therefore, we consider the sample sizes in our study sufficiently large to justify the use of Welch’s t -test.

Response to Reviewer#2

General Comment

This is a nicely written manuscript reporting the gradient distribution of DCS cells along the colon and the underlying role of region specific glycosylation in colon health. The authors elaborate on previously reported research to determine correlation of Core3 glycosylation and proximal mucins, and provide scientific evidence linking it to DCS cell distribution. I have a few minor comments and questions as follows-

Response

We truly appreciate the reviewer’s kind and thoughtful comments highlighting both the significance of our findings and the quality of our manuscript. We are encouraged by this positive evaluation and have carefully addressed the comments and questions raised.

Comment (1)

Since this work is heavily relying on immunostaining, could the authors discuss if an isotope control was included in addition to the negative control, to confirm specificity of primary antibody? This is specifically, in reference to figure 4 (showing the UEA-1 reactivity in the in situ PLA).

Response

We thank the reviewer for raising the importance of including an isotype control to confirm the specificity of antibody-based staining. As for conventional immunostaining experiments (Figs 1, 2, and 3), we included appropriate negative controls by omitting the primary antibodies and lectins to confirm the specificity of the signals. Although we did not perform isotype control staining for each primary antibody used in this study, the observed staining patterns were highly consistent with previously reported or expected expression profiles.

As for PLA, to facilitate the interpretation of PLA controls, we additionally performed co-immunostaining using the same probe pairs (i.e., lectins and anti-Muc2 antibody) employed in the PLA experiments. These data have been included as Supporting Information (S1 Fig). Accordingly, we have referred to this addition in the manuscript (Line 461-463) and added some sentences (Line 474-483 and 525-527). We have also referred to S1 Fig in the relevant sentence (Line 607) to improve clarity.

To confirm the specificity of the anti-Muc2 antibody and to validate the lectin-based PLA system, we included two types of controls: (i) a positive control using BC2LCN lectin and anti-Muc2 antibody, which resulted in a PLA signal spatially consistent with the merged co-immunostaining signal of the same probe pair (S1A Fig); and (ii) a negative control omitting both primary probes (lectin and antibody), which produced no significant PLA signal (Fig 4C). These results collectively support the specificity of the antibody and the validity of the signals observed in lectin-based PLA.

Consequently, if any non-specific PLA signal were to be present in the UEA-I–based PLA (Fig 4B), it would most likely stem from the UEA-I itself. Regarding the reviewer’s suggestion of using an isotype control antibody in combination with UEA-I lectin, we believe that such an experiment would not effectively address the potential for nonspecific signals arising from the lectin probe. UEA-I is a lectin, not an antibody, and thus has no corresponding isotype control. In this context, omitting the lectin would be serve as a more practical and informative negative control. Moreover, the reactivity of UEA-I observed in co-immunostaining (Fig 2E and S1B Fig) was localized in secretory granules, consistent with previous reports (Imai Y et al. Infect Immun. 2003;71(2):985-90; Liquori GE et al. Acta Histochem. 2012;114(7):723-32.). Thus, we consider the likelihood of nonspecific UEA-I–mediated PLA signal to be minimal.

There is also a possibility that PLA signals may arise from nonspecific interactions between the antibody and UEA-I, a concern inherent to PLA, which depends on the spatial proximity of the probes. If such nonspecific interactions occurred in the UEA-I–based PLA, signals would be expected across all secretory granules, where both probes are present. However, as shown in Fig 4B, no PLA signals were detected in DCS cells in the proximal and middle colon, despite the presence of both Muc2 and UEA-I reactivity (S1B Fig). This observation suggests that nonspecific interactions between the probes are minimal. Taken together, we believe that our study includes sufficient validation of the specificity and signal reliability in the lectin-based PLA system.

Alternatively, had such clear validation not been achieved, a competition assay using a monosaccharide—such as fucose, which inhibits UEA-I binging—and enzymatic digestion—such as α1,2-fucosidase digestion treatment—could have been employed to assess specificity in lectin-based PLA. However, in light of the robust positive and negative controls already in place, we consider further controls to be unnecessary in the present study.

Comment (2)

I am also curious why Adobe photoshop was used for image analysis, instead of other open source software available for scientific image analysis.

Response

We thank the reviewer for the comment. Adobe Photoshop was used solely for image presentation purposes. Specifically, we applied levels adjustment uniformly across each image to enhance the visibility of fluorescence signals and to ensure consistent contrast across figure panels. No measurements, cell counting, or other quantitative analyses were performed using Adobe Photoshop. We have rev

---

## [Editor Report · Decision Letter 1]

Deep crypt secretory cells shape region-specific mucin glycosylation patterns in the mouse colon

PONE-D-25-10344R1

Dear Dr.Daisuke Sugahara,

We’re pleased to inform you that your manuscript has been judged scientifically suitable for publication and will be formally accepted for publication once it meets all outstanding technical requirements.

Kind regards,

Donatella Mentino

Academic Editor

PLOS ONE

Additional Editor Comments (optional):

There are some typos that need correction (the citation for Mastridonato et al. is incorrect, Mastrodonato is correct) and should be updated. Additionally, in the section "Conservation of DCS-like cells across species," references are missing to strengthen the discussion. To enhance this section, relevant citations from aquatic vertebrates can be included, such as studies on freshwater species (e.g., doi.org/10.1016/j.aquaculture.2008.12.013) and saltwater species (e.g., doi:10.1111/jfb.13871). These references provide evidence supporting the presence of mucin-producing cells with distinct glycosylation patterns from a wide range of vertebrates, suggesting that DCS-like cells may be conserved beyond terrestrial mammals. Incorporating these studies can give a more comprehensive overview of the evolutionary conservation of these cells, extending from aquatic vertebrates to land mammals.
---

## [Editor Report · Acceptance letter]

PONE-D-25-10344R1

PLOS ONE

Dear Dr. Sugahara,

I'm pleased to inform you that your manuscript has been deemed suitable for publication in PLOS ONE. Congratulations! Your manuscript is now being handed over to our production team.

Kind regards,

on behalf of

Dr. Donatella Mentino

Academic Editor

PLOS ONE